# Recognition of Serious Infections in the Elderly Visiting the Emergency Department: The Development of a Diagnostic Prediction Model (ROSIE)

**DOI:** 10.3390/geriatrics10030060

**Published:** 2025-04-25

**Authors:** Thomas Struyf, Lisa Powaga, Marc Sabbe, Nicolas Léonard, Ivan Myatchin, Ben Van Calster, Jos Tournoy, Frank Buntinx, Laurens Liesenborghs, Jan Y. Verbakel, Ann Van den Bruel

**Affiliations:** 1Epi-Centre, Department of Public Health and Primary Care, KU Leuven, Kapucijnenvoer 7, 3000 Leuven, Belgium; jan.verbakel@kuleuven.be (J.Y.V.);; 2Department of General Practice, Julius Center for Health Sciences and Primary Care, University Medical Center Utrecht, Universiteitsweg 100, 3584 CG Utrecht, The Netherlands; 3Department of Emergency Medicine, University Hospitals, Herestraat 49, 3000 Leuven, Belgium; marc.sabbe@uzleuven.be; 4Department of Emergency Medicine, AZ Voorkempen Hospital, Oude Liersebaan 4, 2390 Malle, Belgium; 5Department of Emergency Medicine, Heilig Hart Hospital, Naamsestraat 105, 3000 Leuven, Belgium; 6Department of Development and Regeneration, KU Leuven, Herestraat 49, 3000 Leuven, Belgium; 7Department of Public Health and Primary Care, KU Leuven, Kapucijnenvoer 7, 3000 Leuven, Belgium; jos.tournoy@uzleuven.be (J.T.);; 8Department of Geriatric Medicine, University Hospitals, Herestraat 49, 3000 Leuven, Belgium; 9Department of General Practice, Maastricht University, Peter Debyeplein 1, 6229 HA Maastricht, The Netherlands; 10Department of Clinical Sciences, Institute of Tropical Medicine, Kronenburgstraat 43, 2000 Antwerp, Belgium; lliesenborghs@itg.be; 11Department of Microbiology, Immunology and Transplantation, KU Leuven, Herestraat 49, 3000 Leuven, Belgium; 12Nuffield Department of Primary Care Health Sciences, University of Oxford, Woodstock Road, Oxford OX2 6GG, UK

**Keywords:** diagnosis, serious infection, geriatrics, emergency care, prediction model

## Abstract

**Background/Objectives**: Serious infections in older adults are associated with substantial mortality and morbidity. Diagnosis is challenging because of the non-specific presentation and overlap with pre-existing comorbidities. The objective of this study was to develop a clinical prediction model using clinical features and biomarkers to support emergency care physicians in diagnosing serious infections in acutely ill older adults. **Methods**: We conducted a prospective cross-sectional diagnostic study, consecutively including acutely ill patients (≥65 year) presenting to the emergency department. Clinical information and blood samples were collected at inclusion by a trained study nurse. A prediction model for *any serious infection* was developed based on ten candidate predictors that were further reduced to four ad interim using a penalized Firth multivariable logistic regression model. We assessed discrimination and calibration of the model after internal validation using bootstrapping. **Results**: We included 425 participants at three emergency departments, of whom 215 were diagnosed with a serious infection (51%). In the final model, we retained systolic blood pressure, oxygen saturation, and C-reactive protein as predictors. This model had good discriminatory value with an Area Under the Receiver Operating Characteristic (AUROC) curve of 0.82 (95% CI: 0.78 to 0.86) and a calibration slope of 0.96 (95% CI: 0.76 to 1.16) after internal validation. Addition of procalcitonin did not improve the discrimination of the model. **Conclusions**: The ROSIE model uses three predictors that can be easily and quickly measured in the emergency department. It provides good discriminatory power after internal validation. Next steps should include external validation and an impact assessment.

## 1. Introduction

Older adults represent a particularly vulnerable population within the emergency department (ED) setting, experiencing disproportionately high rates of ED visits and a significantly elevated risk of adverse outcomes, including hospitalization, functional decline, and mortality [1,2]. For example, in the United States, ED visit rates are 1.5 times higher in adults aged 75 and older compared with those aged 45–64, with a corresponding increase in subsequent hospitalizations [3]. Infections pose a significant risk, with those aged 65 and above having a 50-fold higher risk of dying from acute infections compared with younger adults [4,5]. This heightened susceptibility is multifactorial, arising from age-related immunosenescence, the burden of comorbidities, and the increased prevalence of frailty [6,7,8]. Additionally, the risk of hospitalization for infections is four times higher in this age group. Early detection of serious infections at the time of admission may improve prognosis [9].

Accurate and timely diagnosis of infections in older adults is a well-recognized clinical challenge [10], which can result in treatment delays and worsened outcomes [11,12,13]. Atypical infection presentations further complicates this challenge in this population. For example, older pneumonia patients often do not present with cough or sputum production, but rather with confusion or sudden decline in function status [14,15]. Overlap with pre-existing symptoms from comorbidities, such as dyspnea from heart failure, complicates the diagnostic assessment even further. Atypical presentations can occur across various types of infections in older adults, making it difficult to differentiate between infection and other common geriatric syndromes [16,17].

The decision to admit or discharge the patient often needs to be taken in a short time frame; ED clinicians must balance the risk of rapidly evolving infections with the risk of hospital admission and unnecessary treatment in a vulnerable population [18]. Using biomarkers may improve the diagnosis of a serious infection [19,20,21,22,23], but diagnostic accuracy studies of good methodological quality are scarce in this older population [24,25,26].

A recent systematic review demonstrated that the evidence base underpinning the diagnosis of serious infections in older adults is small and haphazard [26]. Only two biomarkers, procalcitonin and C-reactive protein, were investigated by more than one study. Moreover, existing diagnostic studies on infections in this population attending the ED are generally of low methodological quality and vary considerably in terms of characteristics of the included patients, leading to high variability in disease prevalence, ranging from 13% to 48% [19,21], mainly caused by different selection criteria and different definitions of the target condition [24]. Moreover, the focus of these studies lies on a small sub-selection of infectious diseases (i.e., pneumonia and sepsis/bacteremia), while several other infectious diseases can also be life-threatening, depending on individual patient characteristics [26].

Clinical prediction models have emerged as valuable tools to aid in clinical decision-making, and several models have been developed to predict the risk of serious infection in adult populations [27,28,29,30]. Nevertheless, a significant gap remains in the availability of prediction models specifically tailored and validated for older adults [26]. The unique clinical characteristics of this population, including the high prevalence of atypical infection presentations and age-related physiological changes, can compromise the applicability and accuracy of models derived from younger adult cohorts [6,7,8].

There is a clear need for improved diagnostic tools to aid in the early recognition of serious infections in older adults in the emergency department setting. Such tools could help clinicians to more accurately and rapidly identify older adults at high risk of serious infection, allowing for timely intervention and potentially improving patient outcomes [11,12,13]. The current evidence base is limited, and existing studies often focus on specific infections or suffer from methodological shortcomings. To address this knowledge gap, this study aimed to develop and internally validate a novel prediction model for diagnosing ‘any serious infection’ in older adults presenting to emergency care. This model, utilizing easily measurable clinical features and rapidly available biomarkers, has the potential to improve diagnostic accuracy and timeliness, ultimately leading to better patient outcomes.

## 2. Materials and Methods

This study is reported in accordance with the Transparent Reporting of a multivariable prediction model for Individual Prognosis or Diagnosis (TRIPOD) statement (Appendix A) [31].

### 2.1. Study Design and Setting

In this prospective cross-sectional diagnostic study, we consecutively recruited acutely ill adults aged ≥65 years who presented to any of three participating EDs during office hours as part of a larger study to develop a prediction model for ambulatory care. The ED sites were added because of slow recruitment in general practice. However, the prevalence of serious infections in both settings proved to be very different, which introduced data heterogeneity that was too large to combine in one analysis. As a result, only the data from the ED are presented here.

The study was conducted over two years (April 2021 to July 2023) at three Belgian emergency departments: University Hospitals Leuven—Gasthuisberg Campus (a tertiary academic center, 1949 beds, 72,097 ED visits in 2022, 18 months of recruitment), AZ Voorkempen Malle (a 250-bed regional hospital, 25,704 ED visits in 2022, 26 months of recruitment), and Heilig Hart Hospital Leuven (a 287-bed regional hospital, 21,295 ED visits in 2022, 1.5 months of recruitment). All hospitals adhere to national and regional infection prevention and control guidelines. More information on the hospital characteristics, the study flow, and interim analysis is presented in Appendix A [32,33,34].

### 2.2. Selection of Participants

We applied the following inclusion criteria: patients had to be ≥65 years, presenting to an ED with an acute illness with a maximum duration of 10 days since onset; there had to be a suspicion of an infectious disease by the treating physician.

We excluded patients who were clinically too unstable to engage in study procedures; patients who had an indwelling catheter in situ; immunocompromised patients or patients receiving medications for systemic immunosuppression, including high-dose corticosteroids (equivalent of >4 mg methylprednisolone); patients who had been hospitalized >24 h in the 7 days before participating or who had undergone surgery in the previous 30 days; and patients who had already been included in the study.

### 2.3. Measurements

Upon first presentation at the ED, the study nurse recorded demographic information and clinical features, including vital signs, the level of confusion (CAM-S short form) [35], the level of consciousness (Glasgow coma scale) [36], and the level of functional independence (KATZ scale) [37]. Other clinical features included red flags for specific infections such as petechial rash or meningeal irritation and clinical features suggestive of potentially serious infections, as noted by ED physicians during their initial examination. At first presentation, venous blood samples were taken for lab-based biomarker tests. Due to the timing, the study nurse was not aware of the biomarker results or the final outcome when assessing the participant. A list of all information collected is available as Appendix A.

We used an electronic case report form (eCRF) on the secured electronic data capture platform Castor EDC [38]. For standardization purposes, study nurses were trained, and all measurements were explained in a separate codebook accessible in the eCRF. The study nurse obtained three venous blood samples at the moment of study inclusion and placed the samples in a time-stamped collection bag, which was then stored in a refrigerator between 2 °C and 8 °C to be collected and analyzed by the designated central laboratory within 6 h (AML lab, Antwerp, Belgium) [39]. The central laboratory ensured user training, result validation, and quality control of their machines. The treating physicians were blinded to the results of the analyses. They were free to take an additional blood sample if they felt this was necessary for their clinical decision-making. If a participant did not consent to give a blood sample, we collected biomarker information from their electronic health records (EHRs), if available. Only routine blood samples collected on day 1, analyzed by the hospital laboratory were considered in that case. Laboratory personnel was blinded to clinical information, including the final outcome.

The following biomarkers were measured: C-reactive protein (CRP), procalcitonin (PCT) and White Blood Cell (WBC) count. CRP was measured using the turbidimetric technique (Architect c16000 analyzer, Abbott, North Chicago, IL, USA). PCT was measured using the Vidas B.R.A.H.M.S. PCT (BioMerieux, Lyon, France). The detection margins for CRP and PCT were 0.1 mg/L and 0.05 ng/mL, respectively.

We collected follow-up information for 30 days after study entry by checking EHRs and by direct patient contact. From the records, we extracted information on comorbidities, immunization status, and chronic medication use at the time of study entry. In the 30 days after study entry, we also collected information on repeat visits, complications, functional decline, mortality, additional testing, hospitalization, repeat visits at the ED, and treatment and diagnostic tests relevant to the illness episode. This follow-up information was used to ascertain the final outcome.

### 2.4. Outcome

The primary outcome of the study was *any serious infection* at presentation, defined as an infection that may have severe consequences such as death or hospitalization. In keeping with a previously developed definition by an international expert panel [40], we considered the following infections as serious: either the patient had died within 30 days after inclusion from possible infection or infection-related consequences; or the patient was hospitalized for more than 24 h while receiving an anti-infectious or supportive treatment for infection-related consequences; or the patient was diagnosed with an infection that was considered a priori as serious, such as pneumonia, pyelonephritis, sepsis, meningitis, appendicitis, and osteomyelitis. Any infection was classified as “serious” when there was documentation of an infection-specific positive reference standard, for example, an infiltrate on chest X-ray for pneumonia (infections that are always considered serious, and their reference standards are listed in Appendix A).

If any of these criteria were not met and there was still doubt as to whether an infection in a particular patient was serious or not, all clinical information collected during the 30-day follow-up was submitted to an expert panel. This panel consisted of three clinical experts from different backgrounds (JT, FB, and LL—specialists in geriatrics, primary care, and infectious diseases, respectively). They were asked to base their decision on a previously created definition of a serious infection in an older patient attending ambulatory care, taking into account all available clinical information, including the presence of functional decline, complications, and the need for close follow-up by the treating physician [40]. In order to avoid incorporation bias, we did not present the biomarker results. All three experts independently assessed the information blinded to the other experts’ assessments, and the final decision was based on the majority of votes.

### 2.5. Statistical Analyses

#### 2.5.1. Predictors and Sample Size Argumentation

Before development of the clinical prediction model, we selected the following ten parameters based on existing evidence and clinical input: age, body temperature, heart rate, respiratory rate, systolic blood pressure, peripheral oxygen saturation, the level of confusion at presentation, and the blood tests CRP, PCT, and an abnormal WBC count.

The initial sample size, using the method by van Smeden et al. [41], was *N* = 900 (Appendix A). A pre-planned interim analysis was scheduled after 300 participants to assess futility of the pre-selected predictors (defined as a univariable AUROC < 0.6) and to monitor the proportion of serious infections. The interim analysis was actually performed after 342 participants (Appendix A). Four predictors were not eliminated (systolic blood pressure, oxygen saturation, CRP, and PCT), and the proportion of serious infections was 45%. Re-estimation of the sample size resulted in *N* = 425.

#### 2.5.2. Development of the Prediction Model

The prediction model was developed using multivariable logistic regression with Firth correction for predicting the presence of any serious infection at presentation. Continuous predictors were modeled using restricted cubic splines with three knots to allow a nonlinear relationship with the logit of the outcome. Backward elimination with alpha 0.1 was performed on the spline terms only. This means that a predictor could not be completely eliminated, only its nonlinear term. Model performance was assessed in relation to discrimination, calibration, and clinical utility.

We performed internal validation using the enhanced bootstrap procedure with 200 bootstrap samples that included the interim analysis and backward elimination steps [42]. Performance measures calculated at internal validation were the AUROC, calibration slope, and net benefit at a range of clinically relevant risk thresholds [43]. A model is suggested in order to lead to clinically useful decisions at a risk threshold if the net benefit of a model is greater than the net benefit of treating all patients and the net benefit of treating none. Threshold probabilities are traditionally defined as the minimum probability of disease at which further intervention, in the form of anti-infective/supportive treatment or additional diagnostic testing, would be warranted. We calculated the net benefit for thresholds between 0.2 and 0.5 for the estimated risk of a serious infection in order to support the decision to initiate anti-infectious treatment when the risk exceeded a certain threshold.

The base case model did not include PCT because PCT was seen by the expert panel as a biomarker of second choice compared with CRP due to its higher cost and its lower availability. In a second exploratory step, we repeated the model development procedure but with the addition of PCT (and its spline term) in order to assess the potential increment in the AUROC [44]. For the multivariable modeling and bootstrap analysis, the % peripheral oxygen saturation was transformed using log(101-O_2_ saturation%), and heart rate, respiratory rate, CRP, and PCT were log2-transformed. Missing data for abnormal WBC counts and PCT were accounted for by using regression imputation without including the outcome, following recent research (Appendix A) [45].

We used SAS software version 9.4 (SAS Institute Inc., Cary, NC, USA) and R version 4.3.2 [46].

## 3. Results

### 3.1. Characteristics of Study Subjects

A total of 425 participants were recruited, of which 320 (78%) were seen at the university hospital ED. The general characteristics of the study population by outcome category are shown in Table 1. The mean age of the study population was 79.5 years, and an equal number of men and women participated.

The outcome was established in 325 participants using the pre-defined rules; the expert panel evaluated the remaining 100 of the 425 participant outcomes (24%). The panel reached a unanimous decision in 46 out of 100 participants (46%), of which 30 were placed in the non-serious category and 16 in the serious category. The overall proportion of participants presenting with a serious infection was 51% (215/425); 54% at the ED of the university hospital (173/320) and 40% at the EDs of the regional hospitals (42/105). Pneumonia was the most common serious infection (44%), followed by sepsis (7%) and infectious acute exacerbation of chronic obstructive pulmonary disease (AECOPD, 7%). In participants without a serious infection, mild to moderate COVID-19 was most common (10%), followed by gastrointestinal infections (8%). Almost a third (27%) of all participants had a diagnosis other than an infection, such as cholecystolithiasis or heart failure. These participants were also analyzed in the group that did not have a serious infection.

During the 30-day follow-up period, 23 participants died (5%), of whom 14 (61%) had a serious infection. Three hundred sixty-one participants were hospitalized (85%), of whom 206 (57%) had a serious infection. Over two-thirds (69%) of all participants were referred by a GP.

### 3.2. ROSIE Prediction Model

The univariable AUROC values are listed in Table 2. Of the continuous predictors, the AUROC was the largest for CRP (0.79, 95% CI: 0.75 to 0.84) and PCT (0.73, 95% CI: 0.68 to 0.78).

The final ROSIE prediction model with systolic blood pressure, peripheral oxygen saturation, CRP, and a spline term for CRP is presented in Table 3. The ROSIE model had an AUROC of 0.82 (95% CI: 0.78 to 0.86) and a calibration slope of 0.96 (95% CI: 0.76 to 1.16) after internal validation. In a sensitivity analysis, adding PCT increased the AUROC from 0.820 to 0.821 (+0.001).

Using the model, a patient with, for example, a blood pressure of 100 mmHg, an oxygen saturation of 92%, and a CRP level of 77 mg/L would have an estimated risk of serious infection of 76%. The calculation of this example is presented in Appendix A.

The decision curve for the ROSIE model suggested clinical utility to identify patients who needed prompt anti-infectious and/or supportive treatment or to identify those who needed more invasive or expensive diagnostics in order to establish the final diagnosis (Figure 1).

## 4. Discussion

This study addressed the critical challenge of accurately and promptly diagnosing serious infections in older adults presenting to the ED. Given the atypical presentations of infections in this population and the associated risks of delayed or missed diagnoses, we developed and internally validated a prediction model for ‘any serious infection’ using easily measurable clinical features and rapidly available biomarkers.

### 4.1. Main Findings

About half (51%) of the acutely ill older patients visiting the ED had a serious infection. The ROSIE model included systolic blood pressure, peripheral oxygen saturation, and C-reactive protein, with an AUROC of 0.82 after internal validation. The decision curve analysis indicated potential clinical utility of the model at reasonable risk thresholds (0.2–0.5) to support the decision to initiate anti-infectious treatment.

### 4.2. Comparison with Other Studies

A Dutch team recently developed a clinical prediction model for sepsis in out-of-hours primary care (TeSD-IT study) [30]. Although not specifically designed for older adults, the median age of the participants was 80 years (IQR = 74–85 years). Similar predictors were pre-selected as in the ROSIE study, with emphasis on vital signs and the biomarkers CRP and procalcitonin. They developed a prediction score using age > 65 years, temperature > 38 °C, systolic blood pressure ≤ 110 mmHg, heart rate > 110 beats/min, peripheral oxygen saturation ≤ 95%, and an altered mental status. An AUROC of 0.80 (95% CI: 0.76 to 0.83) was reported for this model. In contrast to our study, adding biomarkers did not improve the diagnostic accuracy of the score. Participants in this study were pre-selected for a home visit by the out-of-hours telephone triage which may explain the very high prevalence of sepsis in this cohort (42%) and the apparent lack of diagnostic accuracy of biomarkers over and above the clinical presentation.

Prendki and colleagues investigated the value of biomarkers for the diagnosis of pneumonia in hospitalized older adults in addition to clinical features such as cough, tachypnea, and fever—but not peripheral oxygen saturation or blood pressure [47]. The univariable AUROCs for CRP (0.64 vs. 0.79 in the ROSIE cohort) and for procalcitonin (PCT, 0.59 vs. 0.73) were lower than in our study. The study was conducted in hospitalized patients with presumptive pneumonia and a prevalence of 67%, which suggests that a high level of pre-selection likely lowered the diagnostic accuracy of any clinical features or biomarkers over and above the clinical evaluation that had already taken place.

### 4.3. Strengths

At present, the ROSIE study is unique in that we developed a prediction model for all serious infections rather than for one specific infection. Once the patient is identified as having an increased risk of a serious infection, appropriate management can be initiated immediately such as applying more invasive or expensive diagnostics or prompt treatment.

We applied a rigorous methodology by consecutively enrolling eligible participants, by assessing only pre-selected candidate predictors, and by developing a parsimonious and adequately powered prediction model that was internally validated. We minimized incorporation bias by assessing the final outcome blinded to the biomarker results.

### 4.4. Limitations

We used an expert panel to decide whether an infection was serious or not in a particular participant when in doubt after applying a set of pre-established criteria. This panel received all clinical information, which inevitably lead to incorporation bias, as the clinical information included predictors such as blood pressure or age. However, they did not receive any biomarker information, and since CRP is the most important predictor in the model, we expected the extent of incorporation bias to be low. We could have lowered this risk of bias by giving them less information; however, this would have increased the potential for misclassification of the outcome and would have defeated the purpose of the expert panel. However, using a panel to determine the outcome of a subset of the study population did lead to differential verification bias, as not all outcomes (serious infection yes/no) were determined with the same information, i.e., using the same reference standard.

We recognize the limitations related to mimicking clinical practice and generalizability. We did not try to standardize all criteria for the specific clinical diagnoses of all infection types, leaving the decision to label a particular infection (e.g., pneumonia) to the treating physician, which may have led to some variability. Furthermore, the inclusion criterion of “a suspicion of an infectious disease”, while reflecting real-world clinical practice where the initial suspicion of infection can be broad (and was clarified for ED physicians as “any situation where an infection (minor or major) is in your differential diagnosis” and emphasizing a broad interpretation), introduces a degree of subjectivity that may affect the reproducibility of our findings and could lead to variability in participant selection. However, using more specific criteria for each type of infection would have been impractical and difficult to implement, and mimicking clinical practice as closely as possible ensures relevance to practice and generalizability of results [48].

For practical reasons, we only recruited participants during office hours, while older patients visiting the ED at night may have a higher risk of serious infection. This may have led to selection bias. To assess this potential for bias, we recorded the number of older patients presenting during the night shift for a month. We estimated that less than 5% of eligible patients were missed by recruiting only during the day, as most patients who presented at night were either too ill to participate in the study, or they were still in the ED in the morning and, therefore, still eligible to participate.

We excluded patients on immunosuppressant medication, which turned out to be the most common reason for exclusion (38% of all exclusions). This means that the ROSIE prediction model is not applicable to patients on such medication. However, in practice these patients will always be considered at high risk of having a serious infection and would be treated accordingly, making the application of any prediction model redundant.

While our model demonstrates good overall performance in identifying serious infections in older adults visiting the ED, it may not fully address the complexities of atypical infection presentation in frail individuals. In frail older adults, infections can manifest with blunted inflammatory responses, leading to lower levels of traditional markers like CRP. This raises the possibility that our model might miss serious infections in this subgroup, as it relies heavily on CRP as a key predictor.

We acknowledge that our study design did not include a specific analysis to evaluate the model’s performance in patients stratified by frailty status and CRP levels. Future research should investigate the predictive value of our model in the context of frailty and atypical presentations with low inflammatory markers. It would be particularly valuable to explore whether incorporating frailty assessments into the model could improve its accuracy in this vulnerable population.

However, it is important to note that while CRP is an important predictor in our model, it is not the sole determinant. The ROSIE model also incorporates other clinical variables, namely systolic blood pressure and oxygen saturation, which may contribute to the identification of serious infections, even in patients presenting with lower CRP levels. These variables may provide important clinical clues to serious infection, independent of the inflammatory response. Nonetheless, further research is needed to fully elucidate the model’s performance across the spectrum of frailty and inflammatory response in older adults.

### 4.5. Clinical Implications

The results of the ROSIE study may be applied in ED settings similar to ours; however, external validation is needed to assess the model’s performance in diverse ED settings, including those with different patient demographics (e.g., higher or lower prevalence of serious infections) and healthcare systems. In settings where the model’s performance is suboptimal, recalibration may be necessary to optimize its accuracy.

Systolic blood pressure and oxygen saturation can be easily measured at the bedside, and CRP is available either as a point-of-care test producing results within minutes or as a widely available laboratory test producing results within an hour. Consequently, the ROSIE model is able to improve and speed up decision-making at the ED.

The early identification of serious infections using tools like our prediction model has the potential to influence hospital length of stay and discharge planning. Hospitalization, especially when prolonged, poses significant risks for older adults, including hospital-acquired infections, functional decline, and increased morbidity. By aiding in timely diagnosis, our model could contribute to more rapid and appropriate management, potentially facilitating earlier and safer discharge to the patient’s home and improving overall outcomes. While our study primarily focused on the development of the prediction model, the impact on length of stay and discharge is an important area for future research.

Although the AUROC is sufficient for clinical application, adding biomarkers other than CRP may be useful to further improve the ROSIE model. For this purpose, we stored a serum sample from each consenting participant in our Biobank for future analyses.

## 5. Conclusions

Our study demonstrates that the ROSIE model, incorporating three easily measurable predictors (systolic blood pressure, oxygen saturation, and C-reactive protein), provides good discriminative power for identifying serious infections in older adults presenting to the ED. This finding offers a promising tool to aid emergency department clinicians in the challenging task of timely diagnosis of serious infections in this vulnerable population, where atypical presentations and comorbidities often complicate assessment. However, to ensure its reliable integration into clinical practice and maximize its benefit for patient care, several steps are crucial. External validation of the ROSIE model in diverse ED settings is strongly recommended to confirm its generalizability. Furthermore, research should explore the added value of incorporating novel biomarkers to potentially enhance the model’s performance. Ultimately, following rigorous validation, the ROSIE prediction model has the potential to be integrated into electronic decision support systems within the ED, providing clinicians with readily accessible and reliable support for decision-making in the management of older adults with suspected serious infections.

## Figures and Tables

**Figure 1 geriatrics-10-00060-f001:**
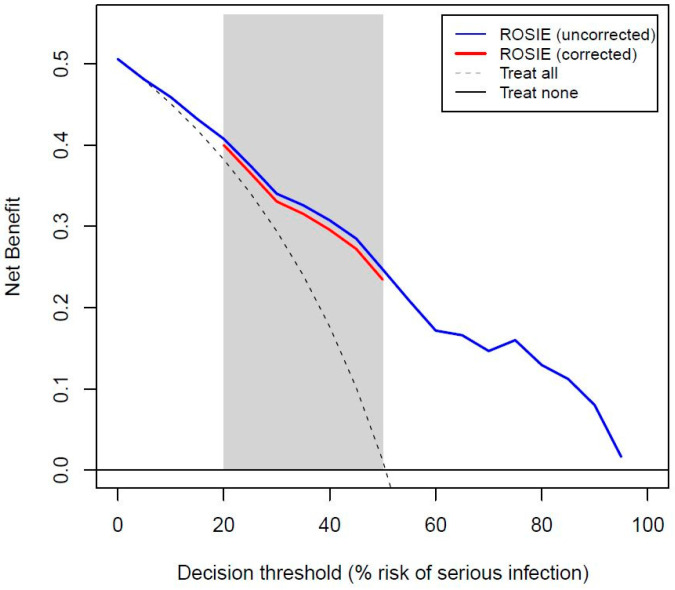
Decision curve for the ROSIE model before and after internal validation.

**Table 1 geriatrics-10-00060-t001:** Population characteristics by outcome.

	Serious Infection (*n* = 215)	No Serious Infection (*n* = 210)	All (*n* = 425)
Age (mean (SD)), years	79.2 (8.2)	79.7 (8.1)	79.5 (8.1)
Age distribution, No. (%)			
65–69 years	31 (14.4)	34 (16.2)	65 (15.3)
70–74 years	42 (19.5)	31 (14.8)	73 (17.2)
75–79 years	47 (21.9)	41 (19.5)	88 (20.7)
80–84 years	36 (16.7)	44 (21.0)	80 (18.8)
85–89 years	36 (16.7)	38 (18.1)	74 (17.4)
90–94 years	19 (8.8)	19 (9.1)	38 (8.9)
95–100 years	4 (1.9)	3 (1.4)	7 (1.7)
Sex, No. (%)			
Women	103 (47.9%)	109 (51.9%)	212 (49.9%)
Final diagnosis, No. (%)			
Pneumonia	94 (43.7)	0 (0.0)	94 (22.1)
Sepsis	16 (7.4)	0 (0.0)	16 (3.8)
Infectious AECOPD	14 (6.5)	0 (0.0)	14 (3.3)
Pyelonephritis/pyelitis	13 (6.1)	0 (0.0)	13 (3.1)
Acute cholecystitis	9 (4.2)	9 (4.3)	18 (4.2)
Bacteremia	8 (3.7)	0 (0.0)	8 (1.9)
COVID-19 *	7 (3.3)	22 (10.5)	29 (6.8)
Acute cystitis	7 (3.3)	13 (6.2)	20 (4.7)
Appendicitis	7 (3.3)	0 (0.0)	7 (1.7)
Acute bronchitis/bronchiolitis	5 (2.3)	7 (3.3)	12 (2.8)
Acute cholangitis	5 (2.3)	1 (0.5)	6 (1.4)
Influenza-like illness	3 (1.4)	3 (1.4)	6 (1.4)
Skin infection	4 (1.9)	6 (2.9)	10 (2.4)
Gastrointestinal infection	6 (2.8)	17 (8.1)	23 (5.4)
Acute prostatitis	1 (0.5)	3 (1.4)	4 (0.9)
Wound infection	1 (0.5)	2 (1.0)	3 (0.7)
Infectious endocarditis	2 (0.9)	0 (0.0)	2 (0.5)
Meningitis	1 (0.5)	0 (0.0)	1 (0.2)
Upper respiratory tract infection	4 (1.9)	5 (2.4)	9 (2.1)
Cellulitis	1 (0.5)	3 (1.4)	4 (0.9)
Non-infectious AECOPD	0 (0.0)	5 (2.4)	5 (1.2)
Viral infection (non-specified)	0 (0.0)	5 (2.4)	5 (1.2)
Infectious arthritis	1 (0.5)	0 (0.0)	1 (0.2)
Other diagnosis	6 (2.8)	109 (51.9)	115 (27.1)
Comorbidities (ten most frequent), No. (%)			
Hypertension	81 (37.7)	90 (42.5)	171 (40.0)
Cholesterolaemia	49 (22.8)	60 (28.3)	109 (25.5)
Cancer (unspecified)	48 (22.3)	36 (17.0)	84 (19.7)
Type II diabetes	37 (17.2)	47 (22.2)	84 (19.7)
Atrial fibrillation	41 (19.1)	34 (16.0)	75 (17.6)
COPD	39 (18.1)	22 (10.4)	61 (14.3)
Heart failure	23 (10.7)	25 (11.8)	48 (11.2)
Chronic kidney disease	19 (8.8)	25 (11.8)	44 (10.3)
Hypothyroidism	16 (7.4)	18 (8.5)	34 (8.0)
Stroke (CVA)	11 (5.1)	16 (7.6)	27 (6.3)
Died during 30-day follow-up, (%)	6.5%	4.3%	5.4%
Hospitalized, (%)	95.8%	73.8%	84.9%
Referral by general practitioner, (%)	66.4%	68.6%	69.1%

* COVID-19 excluding pneumonia: COVID-19 pneumonia was classified as ‘pneumonia’.

**Table 2 geriatrics-10-00060-t002:** Association between diagnostic predictors and serious infection in 425 acutely ill patients presenting to emergency care.

Diagnostic Variable	Missing N(%)	Serious Infection (*n* = 215)	No Serious Infection (*n* = 210)	Univariable AUROC ^1^ with 95% CI	
*Clinical characteristics (Day 1)*					
Age, mean (SD), years	0 (0.0)	79.2 (8.2)	79.7 (8.1)	0.52	0.47 to 0.58
Body temperature, mean (SD), °C	0 (0.0)	37.1 (0.9)	36.9 (0.7)	0.56	0.51 to 0.62
<36.5 °C, N (%)		56 (26.1)	63 (30.0)		
≥38.0 °C, N (%)		39 (18.1)	17 (8.1)		
Heart rate, ^2^ mean (SD), beats/min	0 (0.0)	85.1 (17.6)	82.0 (18.6)	0.56	0.50 to 0.61
Respiratory rate, ^2^ mean (SD), breaths/min	0 (0.0)	19.6 (4.5)	18.4 (3.6)	0.58	0.52 to 0.63
Systolic blood pressure, mean (SD), mmHg	0 (0.0)	131.1 (25.4)	139.7 (24.0)	0.61	0.55 to 0.66
Peripheral oxygen saturation, ^2^ median% (IQR)	0 (0.0)	96.0 (4.0)	97.0 (3.0)	0.67	0.61 to 0.72
Level of confusion (CAM-S score), ^3^	0 (0.0)			0.51	0.45 to 0.56
N (%)					
0		187 (87.0)	186 (88.6)		
1		16 (7.4)	15 (7.1)		
2		4 (1.9)	5 (2.4)		
3		5 (2.3)	1 (0.5)		
4		2 (0.9)	2 (1.0)		
5		0 (0.0)	0 (0.0)		
6		1 (0.5)	1 (0.5)		
*Blood test results (Day 1)*					
C-reactive protein, ^2^ median (IQR), mg/L	0 (0.0)	130.7 (141.9)	38.1 (71.7)	0.79	0.75 to 0.83
Procalcitonin, ^2^ median (IQR), ng/mL	77 (18.1)	0.37	0.03 (0.22)	0.73	0.67 to 0.78
		(1.98)			
Abnormal white blood cell count, ^4^ N (%)	7 (1.6)	86 (40.0)	49 (23.3)	0.58	0.53 to 0.64

^1^ AUROC = Area Under the Receiver Operating Characteristic after imputation of missing values. ^2^ Transformations prior to modeling: heart rate transformed = (log_2_(heart rate)); respiratory rate transformed = (log_2_(respiratory rate)); oxygen saturation transformed = (log(101-O_2_ saturation%)); CRP transformed = (log_2_(CRP+1)); PCT transformed = (log_2_(PCT+1)); prior to transformation, CRP levels < 0.1 mg/L were put to 0.05 mg/L, and PCT levels < 0.05 ng/mL were put to 0.025 ng/mL. ^3^ CAM-S: Confusion Assessment Method—Short form. ^4^ Abnormal white blood cell count: <4000 or >12,000 cells/mm^3^.

**Table 3 geriatrics-10-00060-t003:** The ROSIE model coefficients.

Term in ROSIE Model	Coefficient (95% CI)
Intercept	−2.3898 (−4.2661 to −0.5135)
Systolic blood pressure (mmHg)	−0.0105 (−0.0202 to −0.0011)
Log(101—Peripheral oxygen saturation)	0.9652 (0.6089 to 1.3398)
Log2(C-reactive protein in mg/L)	0.2792 (0.0080 to 0.5854)
Spline term for Log2(C-reactive protein)	0.3782 (0.0798 to 0.6753)

The spline term is calculated as (max(0,(x−2.315868)^3^ − max(0,(x−6.195741)^3^ ∗ (7.939331 − 2.315868)/(7.939331 − 6.195741) + max(0,(x−7.939331)^3^ ∗ (6.195741 − 2.315868)/(7.939331 − 6.195741))/(7.939331 − 2.315868)^2^, where x is the log2(CRP) value.

## Data Availability

Data can be made available upon reasonable request. R-code is available as Appendix A.

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
