# Peer review of "Recognition of Serious Infections in the Elderly Visiting the Emergency Department: The Development of a Diagnostic Prediction Model (ROSIE)"

_geriatrics, 2025, doi:10.3390/geriatrics10030060_

Round 1
Reviewer 1 Report
Comments and Suggestions for Authors
Struyf and colleagues conducted a prospective cross-sectional study to develop a diagnostic prediction model aimed at identifying patients with any serious infection. The study included 452 patients, of whom 215 were ultimately diagnosed with a serious infection. The final model incorporated SBP, O2 sat and CRP. Interestingly, the addition of PCT did not enhance the model's performance.
The study is well written and methodologically sound. However, I have several comments
Major:
1 I have concerns regarding the use of ‘any serious infection’ as the primary outcome. Although the examples provided suggest a clinically meaningful threshold, it is precisely in frail older adults that even infections initially deemed "not serious" can lead to adverse health outcomes. Please elaborate
2 I was surprised to see that infections such as gastroenteritis, COVID-19, and acute cholecystitis/cholangitis were classified as non-serious. While these might be considered mild from an infectious disease standpoint, gastroenteritis, for instance, can rapidly lead to dehydration, acute kidney injury, and a cascade of complications in older patients. This raises concerns about the validity of the binary classification underpinning the model. Although I understand that the expert panel was blinded to biomarker data, the risk remains that the foundational categorization introduced bias into the model, raising the question: if the outcome variable is flawed, are we building on "garbage in, garbage out"?
In line: 4.3% of patients with no serious infection died within 30 days and 74% were hospitalized.
3 While the results and resulting model are plausible and in line with clinical expectations, they are not particularly novel. More importantly, one might question whether the model addresses the most clinically relevant question. In frail older adults, who often present atypically, infections may manifest with low inflammatory markers such as CRP. I would be interested to know whether a subset of patients with low CRP nevertheless had serious infections and worse outcomes, potentially due to frailty-related mechanisms or delayed diagnosis. If this analysis cannot be performed, the authors should at least discuss this
Minor:
1 Pleae refrain from ‘elderly’ which is considered ageist. => older adults/patients/individuals is preferred .
2 I am not sure what Figure 1 adds. If the authors wish to present age distribution, this information could be more appropriately included in Table 1 (with corresponding percentages)
Author Response
Response to comments by reviewer 1
Struyf and colleagues conducted a prospective cross-sectional study to develop a diagnostic prediction model aimed at identifying patients with any serious infection. The study included 452 patients, of whom 215 were ultimately diagnosed with a serious infection. The final model incorporated SBP, O2 sat and CRP. Interestingly, the addition of PCT did not enhance the model's performance.
The study is well written and methodologically sound. However, I have several comments
Major
Comment 1
I have concerns regarding the use of ‘any serious infection’ as the primary outcome. Although the examples provided suggest a clinically meaningful threshold, it is precisely in frail older adults that even infections initially deemed "not serious" can lead to adverse health outcomes. Please elaborate
Response
We appreciate the reviewer's concern regarding the use of ‘any serious infection’ as the primary outcome. We acknowledge the concern that infections initially deemed "not serious" can indeed lead to adverse outcomes, especially in frail older adults.
We would like to elaborate on how we addressed this issue in our study design:
- We defined ‘serious infection’ using pre-established criteria, including documentation of infection-specific positive reference standards (as detailed in Appendix D), death within 30 days due to possible infection, or hospitalization for more than 24 hours with anti-infectious or supportive treatment. These criteria were designed to capture infections with significant clinical consequences.
Our definition of serious infections was based on a previously developed definition by an expert panel of emergency care physicians, general practitioners, geriatricians and infectious diseases specialists. It emphasizes the negative consequences of an infection rather than classifying infections themselves as serious or non-serious. For that reason, if an apparently non-serious infection such as gastro-enteritis resulted in hospitalization for at least 24 hours or death, it was classified as serious.
We added more detail about this in the methods section (subsection ‘outcome’, changes were highlighted).
- For cases where the seriousness of an infection was unclear based on these criteria, we employed an expert panel review. This panel, consisting of specialists in geriatrics, primary care, and infectious diseases, assessed all available clinical information, including functional decline and complications, to make a determination.
- This approach allowed us to include infections that, while perhaps not immediately life-threatening and non-serious at first sight, had the potential to cause significant morbidity and increased healthcare utilization in this vulnerable population.
- We agree that the spectrum of infection severity is particularly important in older adults, and we believe our definition and assessment methods appropriately capture the clinically relevant threshold for serious infection in this population.
Comment 2
I was surprised to see that infections such as gastroenteritis, COVID-19, and acute cholecystitis/cholangitis were classified as non-serious. While these might be considered mild from an infectious disease standpoint, gastroenteritis, for instance, can rapidly lead to dehydration, acute kidney injury, and a cascade of complications in older patients. This raises concerns about the validity of the binary classification underpinning the model. Although I understand that the expert panel was blinded to biomarker data, the risk remains that the foundational categorization introduced bias into the model, raising the question: if the outcome variable is flawed, are we building on "garbage in, garbage out"?
In line: 4.3% of patients with no serious infection died within 30 days and 74% were hospitalized.
Response
We acknowledge that classifying conditions like gastroenteritis, COVID-19, and acute cholecystitis/cholangitis as non-serious may seem counterintuitive, especially in the context of frail older adults.
Our classification was based on a combination of pre-established criteria and expert panel review.
- In the case of gastroenteritis, for example, while we recognize the potential for rapid deterioration in older adults, these cases were classified as non-serious if they resolved without meeting our criteria for serious infection, which included hospitalization for more than 24 hours with anti-infectious or supportive treatment, death within 30 days, or documentation of infection-specific positive reference standards. The occurrence of these types of infections provided an important rationale for a 30-day follow-up period, enabling the collection of all clinical information deemed important to ascertain the final diagnosis and classification.
- The expert panel played a crucial role in these classifications. They were tasked with reviewing cases where the seriousness of the infection was unclear, using a predefined definition of serious infection in older adults. While the panel was blinded to biomarker data, as the reviewer notes, their clinical judgment was essential in determining the ultimate classification.
- We aimed to mitigate the risk of bias by having a diverse panel of experts (geriatrics, primary care, and infectious diseases) to ensure a comprehensive evaluation of each case.
- The 4.3% of patients with no serious infection who died within 30 days died of non-infectious causes, in line with our definition of a serious infection which included death and therefore would have classified the infection as serious if this would have been the cause of death.
We believe that our approach, while not without limitations, represents a rigorous attempt to define and classify serious infections in a way that is clinically relevant to older adults in the emergency department setting. The limitations of using an expert panel were explained in the discussion section.
Comment 3
While the results and resulting model are plausible and in line with clinical expectations, they are not particularly novel. More importantly, one might question whether the model addresses the most clinically relevant question. In frail older adults, who often present atypically, infections may manifest with low inflammatory markers such as CRP. I would be interested to know whether a subset of patients with low CRP nevertheless had serious infections and worse outcomes, potentially due to frailty-related mechanisms or delayed diagnosis. If this analysis cannot be performed, the authors should at least discuss this.
Response
We appreciate the reviewer's comment on the clinical relevance of our model in frail older adults. We agree that this population can present with serious infections despite having low inflammatory markers like CRP, and that this is a crucial consideration.
In response to the reviewer's concern:
- We recognize that our model, while performing well, uses predictors that are already established in clinical practice. Our aim, however, was to develop a simple and easily applicable tool for a complex and vulnerable population in the emergency department setting, where rapid decision-making is essential. The value of our model lies in its ability to combine these predictors in a robust and validated way.
- The reviewer raises a very important point about frail older adults and low CRP. Unfortunately, we did not specifically analyze a subset of patients with low CRP and serious infections with worse outcomes. Our study design did not prospectively stratify patients based on frailty scores, which would be needed to perform such an analysis. However, we did capture data on functional independence using the Katz scale, which could be used in future research to explore this.
- However, we conducted some post-hoc descriptive analyses. While a formal analysis including frailty-related mechanisms could not be performed, we examined the data to see if there were instances of patients with low CRP who had serious infections (false negatives). When using a low cut-off value for CRP (≤5mg/l), the number of false negatives was very low (n=5). Only 2.3% of people with a serious infection had a low CRP and the sensitivity of CRP for the diagnosis of a serious infection was 97.7% at this cut-off.
At a higher cut-off (≤10mg/l), the number of false negatives was still low (n=7, 3.3% of people with a serious infection).
A review of the five individuals with low CRP (≤5 mg/L) revealed the following:
- One 93-year-old male (CRP 2.7 mg/L) with an oxygen saturation of 85% was hospitalized with pneumonia.
- One 78-year-old male (CRP 5.0 mg/L) with a systolic blood pressure of 95 mmHg, a temperature of 38.0°C, and normal oxygen saturation was hospitalized with urosepsis.
- Three patients (91-year-old female, 80-year-old male, and 68-year-old female, with CRP levels of 2.0, 3.8, and 4.7 mg/L, respectively) had oxygen saturations of 93%, 95%, and 93% and were hospitalized with pneumonia.
In conclusion, even in patients with low CRP, serious infections would be identifiable due to a combination of factors: the inclusion of systolic blood pressure and oxygen saturation in the ROSIE model, and the patients' clinical presentation, including signs, symptoms, and comorbidities.
- In the discussion, we have added a more thorough discussion of the limitations of our study in addressing this specific issue (highlighted). We emphasized the need for future research to investigate the predictive value of our model in the context of frailty and atypical presentations with low inflammatory markers. We also highlighted that while CRP is an important predictor, the model also incorporates other clinical variables (systolic blood pressure and oxygen saturation) that may help identify serious infections even in patients with low CRP.
Minor
Comment 1
Please refrain from ‘elderly’ which is considered ageist. => older adults/patients/individuals is preferred .
Response:
We understand the concern regarding the term ‘elderly’ and its potential to be perceived as ageist. We have aimed to avoid this term in the manuscript. However, ‘elderly’ is incorporated into the ROSIE acronym, and our use of the term is limited to instances where we refer directly to the acronym.
Comment 2
I am not sure what Figure 1 adds. If the authors wish to present age distribution, this information could be more appropriately included in Table 1 (with corresponding percentages)
Response
We agree with the reviewer that the age distribution information can be effectively presented in Table 1. We removed Figure 1 and included the age distribution data, with corresponding percentages, in the revised Table 1.
Reviewer 2 Report
Comments and Suggestions for Authors
This study focuses on a relevant topic and provides relevant findings for older adults' care. The study seems to be well conducted. However, I have some suggestions to improve the quality of the reporting in some parts of the manuscript. There are some subsections that are described accurately, whereas other requires some adjustiments. Please find attached my specific comments.

Author Response
Response to comments by Reviewer 2
This study focuses on a relevant topic and provides relevant findings for older adults' care. The study seems to be well conducted. However, I have some suggestions to improve the quality of the reporting in some parts of the manuscript. There are some subsections that are described accurately, whereas other requires some adjustments. Please find attached my specific comments.
Title
Clear and informative.
Introduction
Comment 1
This section is too concise. Please explain the reported concepts. For example, the first paragraph should be based on the first sentence. This means for example that the authors should report epidemiological data and additional explanations that support the sentence. The same for the rest of the introduction.
Response
We acknowledge that this section could be expanded to provide more detailed background information. However, we deliberately restricted the length of the introduction in favor of a more detailed description of the study’s methods.
Given the complexity of developing and validating a clinical prediction rule, we prioritized a comprehensive and transparent explanation of our methodological approach. This decision was driven by the word count limitations and our belief that a detailed methods section is crucial for reproducibility and for allowing readers to critically appraise the validity and potential limitations of our model.
The included references in the introduction provide further information on the epidemiology and challenges of diagnosing serious infections in older adults, and we believe that the current version provides sufficient context for the reader to understand the study's rationale and objectives. We are, however, open to making targeted expansions to the introduction if the editor deems it necessary, provided that adjustments can be made elsewhere in the manuscript to maintain the overall word count.
We did expand the final paragraph of the introduction section (see next comment).
Comment 2
The last paragraph of the introduction should summarize the thesis statement, knowledge gap, and study novelty. These elements are essential to introduce and justify the study’s aim.
Response
We agree that explicitly stating a clearer summary of the knowledge gap and study novelty in the final paragraph of the introduction would provide a more robust justification for our study's aim.
We revised and expanded the final paragraph as follows:
“In summary, there is a clear need for improved diagnostic tools to aid in the early recognition of serious infections in older adults in the emergency department setting. The current evidence base is limited, and existing studies often focus on specific infections or suffer from methodological shortcomings. To address this knowledge gap, this study aimed to develop and internally validate a novel prediction model for diagnosing ‘any serious infection’ in older adults presenting to emergency care. This model, utilizing easily measurable clinical features and rapidly available biomarkers, has the potential to improve diagnostic accuracy and timeliness, ultimately leading to better patient outcomes.”
We hope the additional word count will be acceptable to the editor.
Materials and Methods
I suggest adding a description of the context, type of hospital, structure, and available policies about infection control and son on. This will help to interpret the study’s findings.
Response
We agree that providing further context regarding the participating hospitals will aid in the interpretation of our findings and improve the manuscript's transparency. We added additional information on the type of hospitals to Appendix B, in order to respect word count limitations.
Comment
I couldn’t find the Appendix B.
Response
All appendices have been submitted via the journal's online system. We resubmitted the revised Appendix B. Should you find that any appendix is unavailable, please contact the editorial office for assistance.
Selection of participants
Comment
The authors reported: “There had to be a suspicion of an infectious disease by the treating physician”. What criteria were established to identify a suspicion of an infectious disease?
Response
In our study, the treating physician's suspicion of an infectious disease was the primary inclusion criterion. This decision was based on the physician's clinical judgment at the initial presentation of the patient in the emergency department.
It is important to acknowledge that this approach reflects real-world clinical practice, where the diagnosis of infection often relies on the physician's clinical experience. While this may introduce some variability, it also enhances the generalizability of our findings. We explained this approach in the limitations section of the discussion.
Comment
What about the consent form? What about the data confidentiality and de-identification?
Response
The informed consent form was submitted as part of the manuscript's supporting documentation through the journal's online submission system, as per the journal's requirements. This form detailed the study's purpose, procedures, potential risks and benefits, and the participants' right to withdraw. A copy is available to the editor for review.
We adhered to strict protocols to ensure data confidentiality and de-identification. These protocols were a requirement of the ethical review committee that approved the study. The measures included:
- Using unique study identifiers
- Storing data on secure, password-protected servers
- Limiting access to the data to authorized research personnel.
- Aggregating data for analysis and reporting to prevent individual identification
- Further details of these procedures are described in the study protocol, which was also registered at Clinicaltrials.gov (ID: NCT04516187)
Results
Comment
Table 1: What does it mean “No serious infection”. Please define this criterion. What about confounding variables?
Response
In Table 1, "No serious infection" refers to patients in whom, after thorough evaluation and follow-up, no serious infection was diagnosed. In short, all those not categorized as “serious infection” (binary outcome).
In diagnostic prediction modeling, the primary goal is to accurately classify or predict the presence or absence of a specific condition or disease. In this context, the focus is on identifying the best predictors that, when combined, can effectively discriminate between individuals with and without the condition of interest.
Including relevant variables in the model and using appropriate statistical techniques was therefore essential to ensure validity. In our multivariable logistic regression analysis, we included preselected variables as covariates to reflect the multivariable nature of clinical diagnosis, rather than to adjust for their potential confounding effects. As we see it, confounding is not a central issue in prediction research: serious infection was determined based on all available information; confounding is not an issue when two groups can be categorized accurately. We believe this approach is appropriate for developing a robust and clinically useful prediction tool.
Comment
What about missing data and how they were handled?
Response
In our study, we encountered missing data for some predictor variables. The extent of missing data for each predictor is detailed in Table 2. The strategy used to address this missing data, including the imputation method, is described in Appendix E.
Discussions
Comment
The first paragraph of the introduction should summarize the study’s rationale briefly.
Response
We have added a first paragraph as an introduction to the discussion to briefly restate the clinical problem, its importance, and how our study aimed to address it. We agree that this will provide a helpful transition from the results to the discussion and remind the reader of the study's context and purpose.
The added first paragraph reads:
“This study addressed the critical challenge of accurately and promptly diagnosing serious infections in older adults presenting to the ED. Given the atypical presentations of infections in this population and the associated risks of delayed or missed diagnoses, we developed and internally validated a prediction model for ‘any serious infection’ using easily measurable clinical features and rapidly available biomarkers.”
Comment
The discussion should reflect any relevant findings as well as it should be also focused on the sample characteristics and setting.
Response
We have added some reflections on the sample characteristics and setting when discussing the findings. The adaptations are highlighted throughout the revised discussion (subsection ‘clinical implications’ and ‘conclusions’).
Comment
I suggest moving the subsections “Strengths” and “Limitations” at the end of the discussion section.
Response
We agree that re-ordering the discussion section enhances its flow. Therefore, we have repositioned the "Strengths" and "Limitations" subsections to appear near the end of the discussion, preceding the "Clinical Implications" section. This placement ensures that the strengths and limitations are considered when interpreting the clinical implications.
Conclusions
Comment
This section is too concise. Please restructure this section that should be written as an inverted introduction (please see the following link: https://www.newcastle.edu.au/ data/assets/pdf_file/0009/333765/LD-Conclusions-LH.pdf )
Response
We have revised the conclusion paragraph to follow the inverted introduction structure, as detailed in the provided link.
The conclusion section now reads:
"Our study demonstrates that the ROSIE model, incorporating three easily measurable predictors (systolic blood pressure, oxygen saturation, and C-reactive protein), provides good discriminative power for identifying serious infections in older adults presenting to the ED. This finding offers a promising tool to aid emergency department clinicians in the challenging task of timely diagnosis of serious infections in this vulnerable population, where atypical presentations and comorbidities often complicate assessment. However, to ensure its reliable integration into clinical practice and maximize its benefit for patient care, several steps are crucial. External validation of the ROSIE model in diverse ED settings is strongly recommended to confirm its generalizability. Furthermore, research should explore the added value of incorporating novel biomarkers to potentially enhance the model's performance. Ultimately, following rigorous validation, the ROSIE prediction model has the potential to be integrated into electronic decision support systems within the ED, providing clinicians with readily accessible and reliable support for decision-making in the management of older adults with suspected serious infections."
Additional remark
We thank the reviewer for all the useful suggestions and comments. These revisions have resulted in exceeding the allowed word count, and we hope the editor understands the added value they provide.
Reviewer 3 Report
Comments and Suggestions for Authors
Dear Authors,
This is an interesting and timely topic. Your study has several strengths, mainly related to the multicenter nature of the investigation, the model’s discrimination performance and calibration following internal validation, the use of bootstrapping, and pre-specified candidate predictors for the internal validation.
I would like to highlight some areas that may strengthen the reporting.
- Introduction: My advice is to strengthen the rationale sustaining the aim of the study.
- Methods: the potential variability in physician-defined diagnoses (e.g., pneumonia) may warrant additional comment, especially with regard to generalizability across health systems in relation to the definition of serious infections.
- General comment/discussion: I suggest adding some clarification regarding the model use. I understand that the predictors are readily accessible in most EDs, but I think it may be helpful to provide more practical guidance on how the ROSIE model could be implemented in practice in a more actionable way, such as how it can be received or applied within a scoring system. In addition, the call for additional external validation is appropriate; however, a concise discussion on the population/settings where the model may perform similarly or differently or where recalibration may be necessary could be added.
Author Response
Response to comments by Reviewer 3
Dear Authors,
This is an interesting and timely topic. Your study has several strengths, mainly related to the multicenter nature of the investigation, the model’s discrimination performance and calibration following internal validation, the use of bootstrapping, and pre-specified candidate predictors for the internal validation.
I would like to highlight some areas that may strengthen the reporting.
- Comment
Introduction: My advice is to strengthen the rationale sustaining the aim of the study.
Response
We agree that explicitly stating a clearer summary of the knowledge gap and study novelty in the final paragraph of the introduction would provide a more robust justification for our study's aim.
We revised and expanded the final paragraph of the introduction as follows:
“In summary, there is a clear need for improved diagnostic tools to aid in the early recognition of serious infections in older adults in the emergency department setting. The current evidence base is limited, and existing studies often focus on specific infections or suffer from methodological shortcomings. To address this knowledge gap, this study aimed to develop and internally validate a novel prediction model for diagnosing ‘any serious infection’ in older adults presenting to emergency care. This model, utilizing easily measurable clinical features and rapidly available biomarkers, has the potential to improve diagnostic accuracy and timeliness, ultimately leading to better patient outcomes.”
We hope the additional word count will be acceptable to the editor.
- Comment
Methods: the potential variability in physician-defined diagnoses (e.g., pneumonia) may warrant additional comment, especially with regard to generalizability across health systems in relation to the definition of serious infections.
Response
Indeed, the decision to label a particular infection as pneumonia, for example, was left to the treating physician, and we agree that this may have led to some variability in clinical diagnoses. However, using more specific criteria for each type of infection would have been impractical and difficult to implement. Moreover, mimicking clinical practice as closely as possible ensures relevance to practice and generalizability of results. We explained this in the limitations section of the discussion.
- Comment
General comment/discussion: I suggest adding some clarification regarding the model use. I understand that the predictors are readily accessible in most EDs, but I think it may be helpful to provide more practical guidance on how the ROSIE model could be implemented in practice in a more actionable way, such as how it can be received or applied within a scoring system. In addition, the call for additional external validation is appropriate; however, a concise discussion on the population/settings where the model may perform similarly or differently or where recalibration may be necessary could be added.
Response
We agree that providing more specific guidance on its potential implementation will enhance the clinical utility of our findings. We have expanded our discussion on external validation to include considerations for population and setting variability and the potential need for recalibration, and we have added some explanation on how the ROSIE model could be integrated into clinical practice. (changes are highlighted in the discussion section: ’clinical implications’ and ‘conclusions’ subsections).
Reviewer 4 Report
Comments and Suggestions for Authors
I consider that this mansucript is of the intrest as it gives a tool that can be very useful in the treatment of acute infections in elderly people.
The work uses a convenient methodology that shows the benefits of using ROSIE.
I have some things to comment:
- One important problem with elderdy patients is that their situation can worse when being in the hosptial. Therefore it would be important to comment the impact that a good and fast diagnostic of infection can have in an early discharge to their home.
- I consider that Figure 1 is not necessary, these results could be commented in the results text.
Author Response
Response to comments by reviewer 4
I consider that this manuscript is of the interest as it gives a tool that can be very useful in the treatment of acute infections in elderly people.
The work uses a convenient methodology that shows the benefits of using ROSIE.
I have some things to comment:
Comment 1
One important problem with elderly patients is that their situation can worse when being in the hospital. Therefore it would be important to comment the impact that a good and fast diagnostic of infection can have in an early discharge to their home.
Response
We agree that this is an important point: hospitalization, particularly prolonged hospitalization, carries considerable risks for older adults, including hospital-acquired infections, functional decline, and increased morbidity. Our prediction model, by aiding in the timely identification of serious infections, could contribute to a more rapid and appropriate management plan. This may lead to a decrease in the length of hospital stay and facilitate earlier and safer discharge to the patient's home, which could improve overall outcomes. While our study focused on the development of the prediction model, we acknowledge the importance of this aspect and we included a discussion of it in the ‘clinical implications’ section (highlighted in the discussion section).
Comment 2
I consider that Figure 1 is not necessary, these results could be commented in the results text.
Response
We agree with the reviewer that Figure 1 is not necessary.
We removed Figure 1 and included the age distribution data, with corresponding percentages, in the revised Table 1 (changes were highlighted).
Round 2
Reviewer 1 Report
Comments and Suggestions for Authors
Thank you for the detailed responses and the revised manuscript. While I do not fully agree with all of the authors’ replies, particularly regarding the panel and aspects of validity, I recognize that complete agreement is not essential in the context of peer review. The authors have now elaborated on this issue in both the Methods and Discussion sections. Overall, the manuscript has improved through the revisions and is, in my view, suitable for publication
Author Response
Reviewer 1
Thank you for the detailed responses and the revised manuscript. While I do not fully agree with all of the authors’ replies, particularly regarding the panel and aspects of validity, I recognize that complete agreement is not essential in the context of peer review. The authors have now elaborated on this issue in both the Methods and Discussion sections. Overall, the manuscript has improved through the revisions and is, in my view, suitable for publication.
Response
Thank you for your thoughtful review of our revised manuscript and for acknowledging the improvements we have made.
We understand and respect your perspective that complete agreement was not reached on all points, particularly regarding the panel and aspects of validity. We are glad that our elaborations in the Methods and Discussion sections have provided sufficient clarity on our reasoning and approach in these areas.
We are grateful for your constructive feedback throughout the review process, which has undoubtedly helped to strengthen our work.

Reviewer 2 Report
Comments and Suggestions for Authors
My previous comments have not been addressed entirely. Please find below my comments:
- Scientific writing consists of the ability to be concise and simultaneously express the necessary concepts to adhere to the standards of scientific writing. Without epidemiological data, the size of the problem cannot be conveyed, and therefore neither is the need to undertake a study on the subject. Please avoid general sentences and be incisive with necessary information. Then, I suggest that the authors revise it according to my previous comment.
- Please explain what "a suspicion of an infectious disease" meant. This is not a real-world study, therefore, the criteria should be specified to enable the study evaluation and reproducibility. Even in clinical practice, clinicians base their assumptions on specific observations and parameters. This aspect should be described. Further, this limit was not addressed in the limitations section.
- Information about the consent form and data confidentiality (and any other ethical considerations) must be reported in the manuscript for reasons of transparency and rigor toward the readers.
- A brief description of the context should be reported in the manuscript rather than in the Appendix.
- A binary outcome is “infection yes/no”. “Serious infection” implies a grading. This grading should be described.
I understand that some revisions lead to exceeding the word count, but some essential details must be reported in the manuscript. Scientific writing requires a balance between providing enough information and avoiding unnecessary details, selecting distinct words to convey the message effectively. This approach enables to reporting of the necessary information that can not be omitted for word count purposes.
Author Response
Reviewer 2
My previous comments have not been addressed entirely. Please find below my comments:
Comment 1
Scientific writing consists of the ability to be concise and simultaneously express the necessary concepts to adhere to the standards of scientific writing. Without epidemiological data, the size of the problem cannot be conveyed, and therefore neither is the need to undertake a study on the subject. Please avoid general sentences and be incisive with necessary information. Then, I suggest that the authors revise it according to my previous comment.
Response
We thank the reviewer for their valuable feedback and agree that providing more epidemiological context and detailed explanations would strengthen the Introduction. We have revised the Introduction to include additional epidemiological data and further explanations of the concepts presented, incorporating additional references where appropriate. The specific additions or changes are highlighted in the text.
Comment 2
Please explain what "a suspicion of an infectious disease" meant. This is not a real-world study, therefore, the criteria should be specified to enable the study evaluation and reproducibility. Even in clinical practice, clinicians base their assumptions on specific observations and parameters. This aspect should be described. Further, this limit was not addressed in the limitations section.
Response
We thank the reviewer for raising this important point. We agree that the phrasing “a suspicion of an infectious disease” may lead to different interpretations. To clarify this inclusion criterion for the ED physicians, we communicated it as “any situation where an infection (minor or major) is in your differential diagnosis,” emphasizing a broad interpretation.
We acknowledge the reviewer's concern about the need for specific criteria to ensure study evaluation and reproducibility. In response, we wish to highlight that using more specific criteria would have been impractical. It would have resulted in an extensive list of inclusion and exclusion criteria, potentially making the study cumbersome and difficult to conduct in a fast-paced ED environment.
Moreover, a core principle of our research is to closely mimic clinical practice to maintain relevance and generalizability. We are unclear on the reviewer's statement that “this is not a real-world study,” as our explicit aim was to replicate real-life conditions as much as possible. The term “suspicion of an infectious disease” reflects the nuances of clinical decision-making, where physicians often operate under uncertainty.
The term “suspicion of an infectious disease” has the advantage of ecological validity, aligning with real-world clinical practice. However, we recognize the corresponding disadvantage of introducing variability in interpretation, which could indeed impact the reproducibility of our results.
This limitation is now addressed in the limitations section, where it is discussed alongside the related limitation of standardizing specific clinical diagnoses (changes are highlighted in the discussion section):
“We recognize limitations related to mimicking clinical practice and generalizability. We did not try to standardize all criteria for the specific clinical diagnoses of all infection types, leaving the decision to label a particular infection (e.g., pneumonia) to the treating physician, which may have led to some variability. Furthermore, the inclusion criterion of “a suspicion of an infectious disease,” while reflecting real-world clinical practice where the initial suspicion of infection can be broad (and was clarified for ED physicians as “any situation where an infection (minor or major) is in your differential diagnosis,” emphasizing a broad interpretation), introduces a degree of subjectivity that may affect the reproducibility of our findings and could lead to variability in participant selection. However, using more specific criteria for each type of infection would have been impractical and difficult to implement, and mimicking clinical practice as closely as possible ensures relevance to practice and generalizability of results.”
Comment 3
Information about the consent form and data confidentiality (and any other ethical considerations) must be reported in the manuscript for reasons of transparency and rigor toward the readers.
Response
We agree, and we added additional information to the ‘Informed Consent Statement’ and the ‘Institutional Review Board Statement’ below the discussion/conclusions section, as per the journal's requirements, with new additions highlighted.
Comment 4
A brief description of the context should be reported in the manuscript rather than in the Appendix.
Response
We agree that a brief description of the hospital context is valuable for interpreting the study findings. While we provided detailed information in Appendix B, we acknowledge that including a summary in the main manuscript would improve readability and accessibility. We have added a concise description of the study setting to the Methods section (subsection ‘Study design and setting’). Changes were highlighted in the text.
Comment 5
A binary outcome is “infection yes/no”. “Serious infection” implies a grading. This grading should be described.
Response
We respectfully disagree with the reviewer's comment that “serious infection” cannot be a binary outcome. While the term “serious infection” implies a degree of severity in common language, it can be appropriately used as a binary outcome when clearly defined within the study protocol.
In our study, we provided a detailed definition of “serious infection” in the Methods section (Outcome) to ensure clarity and avoid ambiguity. This definition included specific criteria such as:
- Death within 30 days due to infection
- Hospitalization for more than 24 hours with anti-infectious or supportive treatment
- Diagnosis of specific infections considered a priori as serious (e.g., pneumonia, sepsis)
We believe that this explicit definition allowed for a consistent classification of patients into either the “serious infection” or “no serious infection” group, thus justifying its use as a binary outcome.
We would be happy to further clarify this definition in the manuscript if the reviewer deems it necessary.
Additional remark
I understand that some revisions lead to exceeding the word count, but some essential details must be reported in the manuscript. Scientific writing requires a balance between providing enough information and avoiding unnecessary details, selecting distinct words to convey the message effectively. This approach enables to reporting of the necessary information that cannot be omitted for word count purposes.
Response
We acknowledge the reviewer's remark and agree that balancing brevity with the inclusion of all necessary information is a critical consideration. In some cases, providing sufficient detail warrants exceeding the word count.
Reviewer 3 Report
Comments and Suggestions for Authors
Thank you for the updated revision.
Overall, the edits are satisfactory. I still have a comment on the introduction. The edits in the introduction, in my opinion, do not fully address my comment: The sentences have to be supported by literature and the rationale (gaps and significance of addressing gaps) should be better articulated and reported considering the current literature landscape.
Author Response
Reviewer 3
Comments
Thank you for the updated revision.
Overall, the edits are satisfactory. I still have a comment on the introduction. The edits in the introduction, in my opinion, do not fully address my comment: The sentences have to be supported by literature and the rationale (gaps and significance of addressing gaps) should be better articulated and reported considering the current literature landscape.
Response
We thank the reviewer for their continued constructive criticism and we acknowledge that further refinement of the introduction section is needed to explicitly highlight the gaps in current knowledge and the significance of our work in addressing those gaps. We have carefully reviewed the Introduction again and propose revisions, highlighted in the text, to more effectively contextualize our research question and its contribution to the field. We added additional references were needed.
Reviewer 4 Report
Comments and Suggestions for Authors
I consider that the authors have answered the questions expressed in the review report.
Comments on the Quality of English LanguageThe English Language is correct
Author Response
Reviewer 4
Comment
I consider that the authors have answered the questions expressed in the review report.
Response
Thank you very much for taking the time to review our revised manuscript and for confirming that we have adequately addressed the questions raised in your initial report. We appreciate your positive assessment and are pleased that our responses have been satisfactory.

Round 3
Reviewer 2 Report
Comments and Suggestions for Authors
The authors have now addressed all my concerns. Although there are some study limitations regarding how some data were categorized and interpreted, these limitations have been discussed in the limitations section.
Reviewer 3 Report
Comments and Suggestions for Authors
My previous comments have been addressed. I still have one advice: in the last paragraph of the introduction, it is very weird starting with "in conclusion", please revise.